# PROTO-CLIP: A VISION-LANGUAGE PROTOTYPE ALIGNMENT APPROACH FOR FEW-SHOT LEARNING

## ABSTRACT

We propose a novel framework for few-shot learning by leveraging large-scale vision-language models such as CLIP (Radford et al., 2021). Motivated by uni-modal prototypical networks for few-shot learning, we introduce PROTO-CLIP that utilizes image prototypes and text prototypes for few-shot learning. Specifically, PROTO-CLIP adapts the image and text encoder embeddings from CLIP in a joint fashion using few-shot examples. The embeddings from the two encoders are used to compute the respective prototypes of image classes for classification. During adaptation, we propose aligning the image and text prototypes of the corresponding classes. Such alignment is beneficial for few-shot classification due to the reinforced contributions from both types of prototypes. PROTO-CLIP has both training-free and fine-tuned variants. We demonstrate the effectiveness of our method by conducting experiments on benchmark datasets for few-shot learning, as well as in the real world for robot perception. Code will be released upon acceptance.

## 1 INTRODUCTION

We believe that few-shot learning (Wang et al., 2020) is a promising paradigm to enable autonomous machines, such as robots, to recognize a large number of objects. The appeal lies in the ease of data collection—just a few example images is sufficient for teaching a robot a novel object. On the contrary, object model-based approaches build 3D models of objects and then use these 3D models (Calli et al., 2015) for object recognition. Object category-based approaches focus on recognizing category labels of objects such as 80 categories in the MSCOCO dataset (Lin et al., 2014). The limitation of model-based object recognition is the difficulty of obtaining a large number of 3D models for many objects in the real world. Current 3D scanning techniques cannot deal well with metal objects (e.g., knife) or transparent objects (e.g., glass cup). For category-based object recognition, it is difficult to obtain a large number of images for each category in robotic settings. Large-scale datasets for object categories such as ImageNet (Deng et al., 2009) and Visual Genome (Krishna et al., 2017) are collected from the Internet. These Internet images are not very suitable for learning object representations for robot manipulation due to domain differences. Thus, if a robot can learn to recognize a new object from a few images of the object, it is likely to scale up the number of objects that the robot can recognize overcoming the limitations of model-based and category-based object recognition.

The main challenge in few-shot learning is how to achieve generalization with very limited training examples. Learning good visual representations is the key to achieve good performance in few-shot learning (Tian et al., 2020). Although the Internet images are quite different from robot manipulation settings, they can be used to learn powerful visual representations. Recently, the CLIP (Contrastive Language–Image Pre-training) model (Radford et al., 2021) trained with a large number of image-text pairs from the Internet achieves promising *zero-shot* image recognition performance. Using the visual and language representations from CLIP, several few-shot learning approaches (Zhou et al., 2022; Gao et al., 2021; Zhang et al., 2022) are proposed to improve the zero-shot CLIP model. Gao et al. (2021); Zhang et al. (2022) adapt the CLIP image encoder to learn better feature representations, while Zhou et al. (2022) learns prompts for the CLIP model. On the other hand, few-shot learning approaches are studied in the meta-learning framework (Finn et al., 2017). These approaches are aimed at generalizing to novel classes after training. A notable method is Prototypical Network (Snell et al., 2017) and its variants (Triantafillou et al., 2019; Doersch et al., 2020), which demonstrate

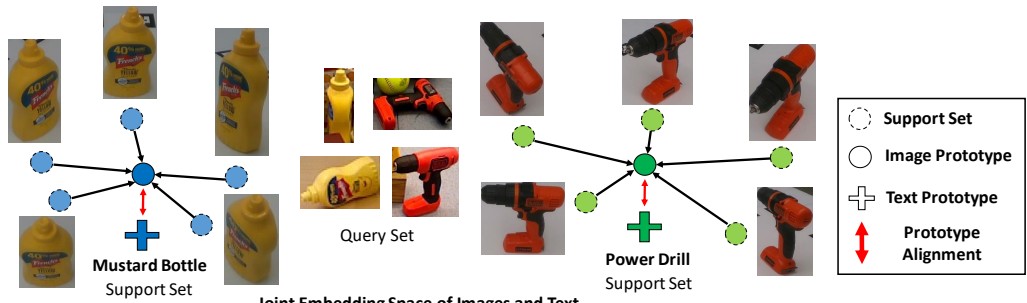

Figure 1: Our PROTO-CLIP model learns a joint embedding space of images and text, where image prototypes and text prototypes are learned and aligned using support sets for few-shot classification.

effective performance for few-shot learning. However, these methods do not leverage the powerful feature representation of CLIP.

These observations motivate us to leverage CLIP in prototypical networks for few-shot learning. We notice that existing methods for adapting CLIP models in few-shot learning adapt the image encoder (Gao et al., 2021; Zhang et al., 2022) or the text encoder (Zhou et al., 2022) in CLIP. We argue that if we can use both the image encoder and the text encoder for classification and jointly adapt the visual and textual features using few-shot training images and prompts, we can improve the few-shot classification performance. To achieve this goal, we propose PROTO-CLIP, a new model motivated by the traditional unimodal Prototypical Networks (Snell et al., 2017). PROTO-CLIP utilizes image prototypes and text prototypes computed from CLIP encoders for classification. In addition, we propose to align the image prototype and the text prototype of the same class during adaptation. In this way, both the image encoder and the text encoder can contribute to the classification while achieving agreement between their predictions. Fig. 1 illustrates the concept of learning the joint embedding space of images and text from PROTO-CLIP.

To verify the effectiveness of PROTO-CLIP, we have conducted experiments on commonly used benchmarks for few-shot image classification, as well as the FEWSOL (P et al., 2023) dataset introduced for few-shot object learning in robotic environments. In addition, we have built a robotic system that integrates Automatic Speech Recognition (ASR), few-shot object recognition using PROTO-CLIP and robotic grasping to demonstrate the robotic application of PROTO-CLIP.

## 2 RELATED WORK

In the context of image recognition, few-shot learning indicates using a few images per image category. The problem is usually formulated as "$N$-way, $K$-shot", i.e., $N$ classes with $K$ images per class. In the traditional image classification setup, these $NK$ images are used as training images. Once a model is trained, it can be used to test images among $N$ classes. Recent CLIP-based few-shot learning methods fall into this setting.

**CLIP-based Few-Shot Learning.** The CLIP (Radford et al., 2021) model applies contrastive learning to image-text pairs from the Internet. It consists of an image encoder and a text encoder for the extraction of features from images and text, respectively. Its training objective is to maximize the similarity between the corresponding image and text in a pair in a high-dimensional joint feature space. After training, CLIP can be used for zero-shot image classification by comparing image features with text embeddings of novel class names. This model is denoted as zero-shot CLIP. When a few training images are available for each class, several approaches are proposed to improve zero-shot CLIP. The linear-probe CLIP model (Radford et al., 2021) trains a logistic regression classifier using CLIP image features. CoOp (Zhou et al., 2022) proposes to use learnable vectors as a prompt for the CLIP text encoder for few-shot learning. CLIP-Adapter (Gao et al., 2021) learns two layers of linear transformations on top of the image encoder and the text encoder with residual connections, respectively, to adapt CLIP features for few-shot learning. Tip-Adapter (Zhang et al., 2022) builds a key-value cache model, where keys are CLIP image features and values are one-hot vectors of the class labels. Given a query image, its image feature is compared with the cache keys to combine the

| Method | Use Support Sets | Adapt Image Embedding | Adapt Text Embedding | Align Image and Text |
|---|:---:|:---:|:---:|:---:|
| Zero-shot CLIP (Radford et al., 2021) | ✗ | ✗ | ✗ | ✓ |
| Linear-probe CLIP (Radford et al., 2021) | ✓ | ✓ | ✗ | ✗ |
| CLIP-Adapter (Gao et al., 2021) | ✓ | ✓ | ✓ | ✗ |
| CoOp (Zhou et al., 2022) | ✓ | ✗ | ✓ | ✗ |
| Tip-Adapter (Zhang et al., 2022) | ✓ | ✓ | ✗ | ✗ |
| Sus-X (Udandarao et al., 2022) | ✓ | ✓ | ✗ | ✗ |
| **PROTO-CLIP (Ours)** | ✓ | ✓ | ✓ | ✓ |

Table 1: Comparison between our proposed method with existing CLIP-based methods for few-shot learning. "Use Support Sets" indicates if a method uses support training sets for fine-tuning. "Adapt Image/Text Embedding" indicates if a method adapts the image/text embeddings from CLIP. "Align Image and Text" indicates if a method specifically aligns images and corresponding text in the feature space.

value labels for classification. Tip-Adapter can also fine-tune the keys by treating them as learnable parameters, which further improves the few-shot classification accuracy. Sus-X (Udandarao et al., 2022) leverages the power of Stable Diffusion (Rombach et al., 2022) to create support sets and aims to address the issue of uncalibrated intra-modal embedding distances in TIP-Adapter (Zhang et al., 2022) by utilizing inter-modal distances as a connecting mechanism.

Table 1 compares our proposed method with existing CLIP-model-based few-shot learning methods. By using the image prototypes and text prototypes for classification, our method can adapt both the image embeddings and text embeddings from CLIP. In addition, the model aligns the image prototypes and the text prototypes, which serves as a regularization term in adapting the feature embeddings. We empirically verify our model by conducting experiments on benchmark datasets for few-shot learning.

**Meta-learning-based Few-Shot Learning.** In parallel with these efforts to adapt CLIP for few-shot learning, meta-learning-based approaches are also proposed for few-shot learning. While previous CLIP-based models are tested on the same classes in training, the focus here is to learn a model on a set of training classes $\mathcal{C}_{train}$ that can generalize to novel classes $\mathcal{C}_{test}$ in testing. Each class contains a support set and a query set. During training, the class labels for both sets are available. During testing, only the class labels of the support set are available, and the goal is to estimate the labels of the query set. Meta-learning-based approaches train a meta-learner with the training classes $\mathcal{C}_{train}$ that can be adapted to the novel classes $\mathcal{C}_{test}$ using their support sets. Non-episodic approaches use all the data in $\mathcal{C}_{train}$ for training such as $k$-NN and its 'Finetuned' variants (Gidaris & Komodakis, 2018; Qi et al., 2018; Chen et al., 2019; Tian et al., 2020). Episodic approaches construct episodes, i.e., a subset of the training classes, to train the meta-learner. Representative episodic approaches include Prototypical Networks (Snell et al., 2017), Matching Networks (Vinyals et al., 2016), Relation Networks (Sung et al., 2018), Model Agnostic Meta-Learning (MAML) (Finn et al., 2017), Proto-MAML (Triantafillou et al., 2019) and CrossTransformers (Doersch et al., 2020). The Meta-Dataset (Triantafillou et al., 2019) was introduced to benchmark few-shot learning methods in this setting. In this work, we consider training and testing in the same classes following previous CLIP-based few-shot learning methods (Zhou et al., 2022; Gao et al., 2021; Zhang et al., 2022).

## 3 METHOD

We consider the $N$-way $K$-shot classification problem. In few-shot settings, $K \ll N$. The image set with class labels is considered as the *support set*: $\mathcal{S} = \{\mathbf{x}_i^s, y_i^s\}_{i=1}^M$, where $\mathbf{x}_i^s$ denotes a support image, $y_i^s \in \{1, 2, \ldots, N\}$ denotes the class label of the support image, and $M$ is the size of the support set. In $N$-way $K$-shot settings, $M = NK$. The goal of few-shot classification is to classify the *query set* $\mathcal{Q} = \{\mathbf{x}_j^q\}_{j=1}^L$, i.e., $L$ test images without class labels. Specifically, we want to estimate the conditional probability $P(y = k|\mathbf{x}^q, \mathcal{S})$ that models the probability distribution of the class label $y$ given a query image $\mathbf{x}^q$ and the support set $\mathcal{S}$.

**Our PROTO-CLIP model (Fig. 2)**. We propose to leverage both the image encoder and the text encoder in the CLIP model (Radford et al., 2021) to estimate the conditional probability of class label

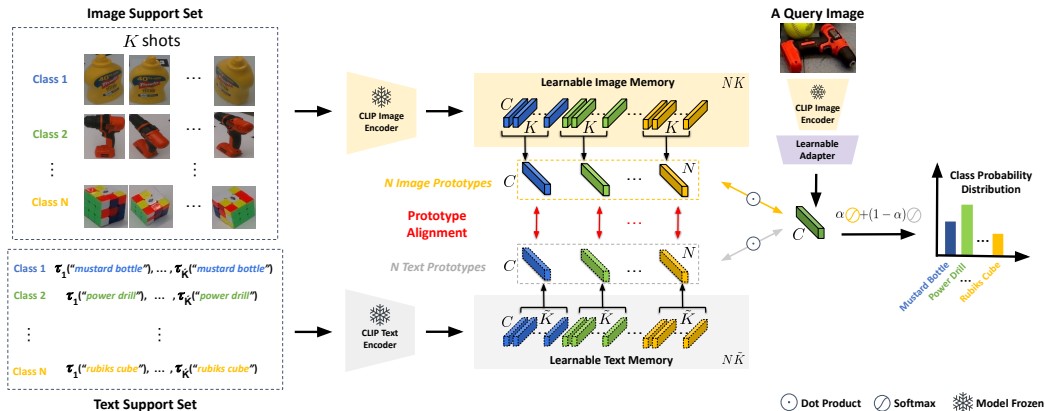

Figure 2: Overview of our proposed PROTO-CLIP model. The CLIP image encoder and text encoder are frozen during training. The image memory, the text memory and the adapter network are learned. Given a class name, $\tau_i$ returns the $i^{th}$ out of $\tilde{K}$ predefined text prompts.

as

$$P(y = k|\mathbf{x}^q, \mathcal{S}) = \alpha \underbrace{P(y = k|\mathbf{x}^q, \mathcal{S}_x)}_{\text{image probability}} + (1 - \alpha) \underbrace{P(y = k|\mathbf{x}^q, \mathcal{S}_y)}_{\text{text probability}}, \tag{1}$$

where $\mathcal{S}_x = \{\mathbf{x}_i^s\}_{i=1}^M$ and $\mathcal{S}_y = \{y_i^s\}_{i=1}^M$ denote the image set and the label set of the support set $\mathcal{S}$, respectively, and $\alpha \in [0, 1]$ is a hyper-parameter to combine the two probabilities. To model the probability distributions conditioned on $\mathcal{S}_x$ or $\mathcal{S}_y$, we leverage the prototypical networks (Snell et al., 2017):

$$P(y = k|\mathbf{x}^q, \mathcal{S}_x) = \frac{\exp(-\beta \|g_{\mathbf{w}_1}(\mathbf{x}^q) - \mathbf{c}_k^x\|_2^2)}{\sum_{k'=1}^N \exp(-\beta \|g_{\mathbf{w}_1}(\mathbf{x}^q) - \mathbf{c}_{k'}^x\|_2^2)}, \tag{2}$$

$$P(y = k|\mathbf{x}^q, \mathcal{S}_y) = \frac{\exp(-\beta \|g_{\mathbf{w}_1}(\mathbf{x}^q) - \mathbf{c}_k^y\|_2^2)}{\sum_{k'=1}^N \exp(-\beta \|g_{\mathbf{w}_1}(\mathbf{x}^q) - \mathbf{c}_{k'}^y\|_2^2)}, \tag{3}$$

where $g_{\mathbf{w}_1}(\cdot)$ denotes the CLIP image encoder plus an adapter network with learnable parameters $\mathbf{w}_1$ used to compute the feature embeddings of query images. The CLIP image encoder is pretrained and then frozen. $\mathbf{c}_k^x$ and $\mathbf{c}_k^y$ are the "prototypes" for class $k$ computed using images and text, respectively. $\beta \in \mathbb{R}^+$ is a hyperparameter to sharpen the probability distributions. We have the prototypes as

$$\mathbf{c}_k^x = \frac{1}{M_k} \sum_{y_i^s = k} \phi_{\text{Image}}(\mathbf{x}_i^s), \quad \mathbf{c}_k^y = \frac{1}{\tilde{M}_k} \sum_{j=1}^{\tilde{M}_k} \phi_{\text{Text}}(\text{Prompt}_j(y_i^s = k)), \tag{4}$$

where $M_k$ is the number of examples with label $k$, and $\tilde{M}_k$ is the number of prompts for label $k$. To compute text embeddings, we can either directly input the class names such as "mug" and "plate" into the text encoder, or convert the class names to phrases such as "a photo of mug" and "a photo of plate" and then input the phrases into the text encoder. These phrases are known as *prompts* of the vision-language models. We can use multiple prompts for each class label. $\phi_{\text{Image}}(\mathbf{x}_i^s)$ and $\phi_{\text{Text}}(\text{Prompt}_j(y_i^s = k))$ denote the image embedding and the $j$th text embedding of the image-label pair $(\mathbf{x}_i^s, y_i^s)$ computed using the CLIP image encoder and the text encoder, respectively. These embeddings with dimension $C$ of the support set form the image memory and the text memory, as shown in Fig. 2. They are learnable embedding vectors initialized by the computed embeddings using the CLIP image encoder and text encoder. We use $\mathbf{c}_k^x$ and $\mathbf{c}_k^y$ to denote the mean of the embeddings of the images and the prompts for class $k$, respectively. Since the image embeddings and the text embeddings are of the same dimension, we can compute the distance between the text prototype $\mathbf{c}_k^y$ and the image embedding $g_{\mathbf{w}_1}(\mathbf{x}^q)$ in Eq. 3. As we can see, our model leverages prototypical networks with image encoder and text encoder from CLIP. We name it "PROTO-CLIP".

**Learning the memories and the adapter.** During training, we can construct a support set $\mathcal{S} = \{\mathbf{x}_i^s, y_i^s\}_{i=1}^M$ and a query set with ground truth labels $\mathcal{Q} = \{\mathbf{x}_j^q, y_j^q\}_{j=1}^L$. Then we can use $\mathcal{S}$ and $\mathcal{Q}$

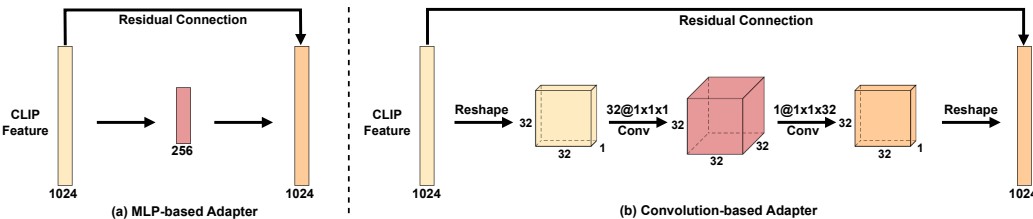

Figure 3: Two designs of the adapters. (a) A Multi-layer perceptron-based adapter as in (Gao et al., 2021). (b) A convolution-based adapter that we introduce. The feature dimension is for CLIP ResNet50 backbone.

to learn the weights in PROTO-CLIP. First, the support set is used to initialize the image memory $\mathbf{W}_{\text{image}}$ and the text memory $\mathbf{W}_{\text{text}}$. Second, the weights in the adapter network applied to the query images $g_{\mathbf{w}_1}(\cdot)$ need to be learned. Fig. 3 shows two designs of the adapter network, i.e., an MLP-based adapter as in (Gao et al., 2021) and a convolution-based adapter that we introduce. The convolution-based adapter has fewer weights to learn compared to the MLP-based one. We found that the two adapters have their own advantages on different datasets in our experiments. Finally, motivated by the CLIP-Adapter (Gao et al., 2021), we do not fine-tune the weights in the image encoder and text encoder by freezing these weights during training. In this way, we can reuse the weights of CLIP trained on a large number of image-text pairs and adapt the image embeddings and the text embeddings.

**Loss Functions.** The first loss function is the negative log-probability of the true label for a query image: $\mathcal{L}_1(\mathbf{W}_{\text{image}}, \mathbf{W}_{\text{text}}, \mathbf{w}_1) = -\log P(y^q = k|\mathbf{x}^q, \mathcal{S})$, where $P(y^q = k|\mathbf{x}^q, \mathcal{S})$ is defined in Eq. 1. Minimizing $\mathcal{L}_1$ learns the weights to classify the query images correctly. Second, we propose aligning the image prototypes and the text prototypes in training. Let $\{\mathbf{c}_1^x, \mathbf{c}_2^x, \ldots, \mathbf{c}_N^x\}$ be the image prototypes computed from the image embeddings for the $N$ classes and $\{\mathbf{c}_1^y, \mathbf{c}_2^y, \ldots, \mathbf{c}_N^y\}$ be the corresponding text prototypes. We would like to learn the model weights such that $\mathbf{c}_k^x$ is close to $\mathbf{c}_k^y$ and far from other prototypes in the embedding space. We utilize the InfoNCE loss for contrastive learning (Oord et al., 2018):

$$\mathcal{L}_2^k(\mathbf{c}_k^x, \{\mathbf{c}_{k'}^y\}_{k'=1}^N) = -\log \frac{\exp(\mathbf{c}_k^x \cdot \mathbf{c}_k^y)}{\sum_{k'=1}^N \exp(\mathbf{c}_k^x \cdot \mathbf{c}_{k'}^y)}, \mathcal{L}_3^k(\mathbf{c}_k^y, \{\mathbf{c}_{k'}^x\}_{k'=1}^N) = -\log \frac{\exp(\mathbf{c}_k^y \cdot \mathbf{c}_k^x)}{\sum_{k'=1}^N \exp(\mathbf{c}_k^y \cdot \mathbf{c}_{k'}^x)}$$
(5)

for $k = 1, \ldots, N$, where $\cdot$ indicates dot-product. Here, $\mathcal{L}_2^k(\mathbf{c}_k^x, \{\mathbf{c}_{k'}^y\}_{k'=1}^N)$ compares an image prototype $\mathbf{c}_k^x$ with the text prototypes $\{\mathbf{c}_{k'}^y\}_{k'=1}^N$, while $\mathcal{L}_3^k(\mathbf{c}_k^y, \{\mathbf{c}_{k'}^x\}_{k'=1}^N)$ compares a text prototype $\mathbf{c}_k^y$ with the image prototypes $\{\mathbf{c}_{k'}^x\}_{k'=1}^N$. In this way, we can align the image prototypes and the text prototypes for the $N$ classes. This alignment can facilitate classification, since the class conditional probabilities are computed using the image prototypes and the text prototypes as in Eqs. 2 and 3. The total loss function for training is:

$$\mathcal{L} = -\frac{1}{L}\sum_{j=1}^L \log P(y_j^q = k|\mathbf{x}_j^q, \mathcal{S}) + \frac{1}{N}\sum_{k=1}^N \left(\mathcal{L}_2^k(\mathbf{c}_k^x, \{\mathbf{c}_{k'}^y\}_{k'=1}^N) + \mathcal{L}_3^k(\mathbf{c}_k^y, \{\mathbf{c}_{k'}^x\}_{k'=1}^N)\right) \quad (6)$$

for a query set $\mathcal{Q} = \{\mathbf{x}_j^q, y_j^q\}_{j=1}^L$. Following previous CLIP-based few-shot learning methods (Zhou et al., 2022; Gao et al., 2021; Zhang et al., 2022), the support set and the query set are the same during training in our experiments, i.e., $\mathcal{S} = \mathcal{Q}$ meaning any of the support samples can act as a query sample during training.

## 4 EXPERIMENTS

**Datasets and Evaluation Metric.** Following previous CLIP-based few-shot learning methods (Zhou et al., 2022; Gao et al., 2021; Zhang et al., 2022), we conduct experiments on the following datasets for evaluation: ImageNet (Deng et al., 2009), StandfordCars (Krause et al., 2013), UCF101 (Soomro et al., 2012), Caltech101 (Fei-Fei et al., 2004), Flowers102 (Nilsback & Zisserman, 2008), SUN397 (Xiao et al., 2010), DTD (Cimpoi et al., 2014), EuroSAT (Helber et al., 2019), FGVCAircraft (Maji et al.,

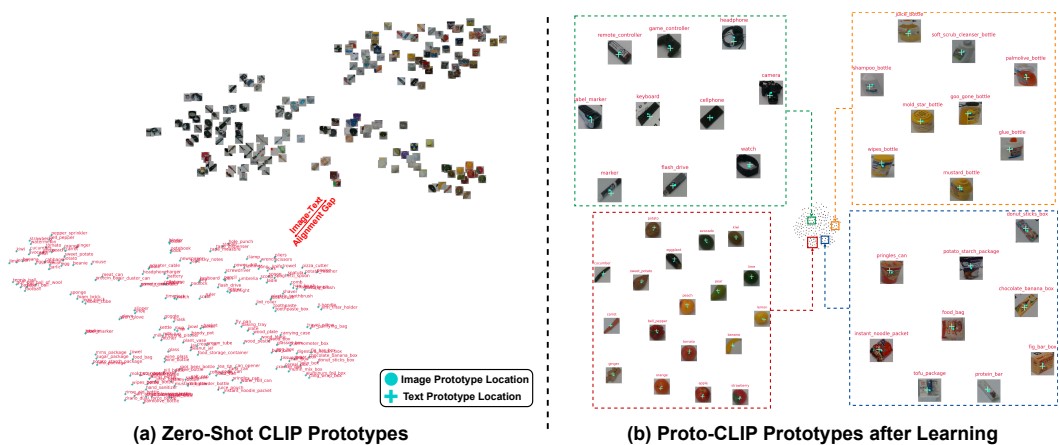

(a) Zero-Shot CLIP Prototypes      (b) Proto-CLIP Prototypes after Learning

Figure 4: Barnes-Hut t-SNE visualization (Van Der Maaten, 2014) using the FewSOL dataset (P et al., 2023). (a) Image and text prototypes from zero-shot CLIP, which are not aligned. (b) Aligned image and text prototypes from PROTO-CLIP-$F$.

2013), OxfordPets (Parkhi et al., 2012), and Food101 (Bossard et al., 2014). In addition, we also include the FewSOL dataset (P et al., 2023) recently introduced for few-shot object recognition in robotic environments in order to improve object classification for robot manipulation tasks. In the $N$-way $K$-shot classification setting, $K$ images for each class will be sampled from each dataset for training. A validation set of each dataset is reserved for hyper-parameter tuning, and a test set is used for evaluation. Following related works, we report the classification accuracy of the test set as the evaluation metric.

**Choosing the Hyper-parameters: $\alpha$ and $\beta$.** From the experiments, we found that the two hyper-parameters $\alpha$ in Eq. 1 and $\beta$ in Eq. 2 and Eq. 3 play a critical role in classification accuracy. Therefore, for each dataset, we conducted a grid search of the two parameters using the validation set. Then we finalize their values for all the runs in our experiments.

**PROTO-CLIP Variants.** i) "PROTO-CLIP": we do not train the image memory and the text memory and do not use any adapter in PROTO-CLIP (Fig. 2), we directly run inference using the pre-trained CLIP features. We term this variant the "training-free" version because it does not require training. This offers a convenient way to quickly test new datasets without the complexities of training, although it comes with the caveat of potential misalignment between visual and textual features. ii) "PROTO-CLIP-$F$": we train the image memory and/or the text memory with the adapter. During training, for all the query images, we precompute their CLIP image features and directly use these stored features for training. This variant can be trained more quickly w.r.t. the following variant. Therefore, we use it for our ablation studies. iii) "PROTO-CLIP-$F$-$Q^T$": During training, for each query image, we apply random data augmentation operations such as cropping and horizontal flip. Then we compute CLIP image features for the transformed query images during training.

## 4.1 ABLATION STUDIES

**Adapter Types and Learnable Text Memory.** Since the 12 datasets have different characteristics, we found that varying adapter types and whether to learn the text memory or not affect performance. Table 2 summarizes the result of this ablation study. Visual data plays a crucial role in image recognition when compared to textual information. Therefore, visual memory keys are consistently trained, regardless of the circumstances. The architectures of the MLP-based adapter and the convolution-based adapter are illustrated in Fig. 3. "2xConv" indicates using 2 convolution layers as shown in Fig. 3, while "3xConv" uses 3 convolution layers in the adapter where we add a $32@3 \times 3 \times 32$ convolution layer in the middle. By checking the best accuracy for each dataset, we can observe that there is no consensus on which adapter and trainable text memory setup to use among these datasets. Therefore, we select the best configuration on the adapter and learnable text memory for each dataset in the following experiments. Learning both image memory and text memory can help to yield aligned image-text prototypes. Fig. 4 visualizes the image-text prototypes in

the FewSOL dataset (P et al., 2023) before and after training. For PROTO-CLIP-$F$, unless specified otherwise, both the adapter and the visual memory keys are trained in all scenarios.

| Adapter | Train-Text-Memory | ImageNet | FGVC | Pets | Cars | EuroSAT | Caltech101 | SUN397 | DTD | Flowers | Food101 | UCF101 | FewSOL |
|---|---|---|---|---|---|---|---|---|---|---|---|---|---|
| MLP | ✗ | 61.06 | 35.31 | 85.61 | 72.19 | 83.47 | 92.58 | 68.54 | 63.89 | 95.01 | 74.05 | 76.16 | 28.65 |
| MLP | ✓ | 61.06 | **37.56** | 85.72 | 73.61 | **83.53** | 92.13 | 69.71 | 63.89 | **96.06** | 74.05 | 76.16 | 32.87 |
| 2xConv | ✗ | **65.75** | 34.38 | **89.62** | 75.25 | 81.85 | 93.40 | **71.94** | 67.85 | 94.76 | **79.09** | **77.50** | 27.13 |
| 2xConv | ✓ | 58.60 | 35.82 | 89.21 | 74.34 | 81.78 | 93.02 | 69.79 | 67.32 | 95.82 | 78.06 | 76.37 | 27.13 |
| 3xConv | ✗ | 65.37 | 34.41 | 88.74 | 75.25 | 82.21 | **93.43** | 71.63 | 67.67 | 94.40 | 79.11 | 77.50 | 29.78 |
| 3xConv | ✓ | 59.63 | 36.15 | 87.93 | 72.68 | 81.57 | 92.74 | 68.64 | **68.56** | 95.78 | 78.61 | 77.03 | **35.22** |

Table 2: Results of ablation study of various query adapter types and textual memory bank training using the CLIP ResNet50 backbone with $K = 16$ on PROTO-CLIP-$F$. In case of a tie, the underlined setup was selected randomly.

**Loss functions.** We have introduced three different loss functions in Sec. 3: $\mathcal{L}_1, \mathcal{L}_2, \mathcal{L}_3$. We analyze the effects of these loss functions in Table 3. We can see that i) the $\mathcal{L}_1$ loss function is essential since it drives the classification of the query images; ii) Overall, both $\mathcal{L}_2$ and $\mathcal{L}_3$ loss functions for prototype alignment contribute to the performance, which verifies our motivation of aligning image and text prototypes for few-shot classification.

| Loss | ImageNet | FGVC | Pets | Cars | EuroSAT | Caltech101 | SUN397 | DTD | Flowers | Food101 | UCF101 | FEWSOL |
|---|---|---|---|---|---|---|---|---|---|---|---|---|
| $\mathcal{L}_1$ | 62.67 | 20.34 | 73.21 | 73.77 | 78.98 | 92.25 | 68.34 | 66.49 | **96.14** | 77.39 | 76.66 | 34.57 |
| $\mathcal{L}_2$ | 62.29 | 4.71 | 0.00 | 0.00 | 38.95 | 0.28 | 66.93 | 67.38 | 10.31 | 77.71 | 57.41 | 32.70 |
| $\mathcal{L}_3$ | 62.27 | 4.14 | 0.00 | 0.00 | 38.09 | 0.24 | 64.86 | 67.38 | 10.27 | 77.69 | 57.55 | 20.22 |
| $\mathcal{L}_1 + \mathcal{L}_2$ | 65.39 | 36.24 | 88.58 | 75.39 | 82.78 | **93.71** | 71.65 | 68.09 | 96.06 | 78.69 | 77.29 | 33.48 |
| $\mathcal{L}_2 + \mathcal{L}_3$ | 62.33 | 3.87 | 0.00 | 0.00 | 36.86 | 0.24 | 64.84 | 68.32 | 8.20 | 77.35 | 57.52 | 19.61 |
| $\mathcal{L}_1 + \mathcal{L}_3$ | 65.43 | 36.84 | 88.58 | **75.51** | 82.84 | 93.35 | 71.44 | 68.32 | **96.14** | 78.80 | **77.53** | 33.43 |
| $\mathcal{L}_1 + \mathcal{L}_2 + \mathcal{L}_3$ | **65.75** | **37.56** | **89.62** | 75.25 | **83.53** | 93.43 | **71.94** | **68.56** | 96.06 | **79.09** | 77.50 | **35.22** |

Table 3: Ablation study of various Loss functions using the CLIP ResNet50 backbone and $K = 16$. The best performing model architectures for each dataset from Table 2 are used here.

**Backbones.** Table 4 shows the results of using different backbone networks on the FewSOL dataset (P et al., 2023). In general, better backbones can learn more powerful feature representations and consequently improve the classification accuracy. CLIP vision transformer backbones achieve better performance than CLIP ResNet backbones.

| Model | Adapter | Train-Text-Memory | Backbone | | | | |
|---|---|---|---|---|---|---|---|
| | | | **RN50** | **RN101** | **ViT-B/16** | **ViT-B/32** | **ViT-L/14** |
| Zero-Shot-CLIP (Radford et al., 2021) | - | - | 25.91 | 32.96 | 40.70 | 41.87 | 54.57 |
| Tip (Zhang et al., 2022) | - | - | 29.74 | 37.43 | 47.00 | 41.48 | 56.78 |
| Tip-F (Zhang et al., 2022) | - | - | 32.52 | 41.43 | 50.17 | 45.48 | 60.17 |
| PROTO-CLIP-$F$ | MLP | ✗ | 33.48 | 39.04 | 47.96 | 41.91 | 58.65 |
| PROTO-CLIP-$F$ | MLP | ✓ | 34.83 | 40.74 | 47.43 | 42.13 | 58.91 |
| PROTO-CLIP-$F$ | 2xConv | ✗ | 35.04 | 41.04 | 50.83 | 46.52 | **63.74** |
| PROTO-CLIP-$F$ | 2xConv | ✓ | 35.04 | 42.52 | 49.26 | 43.43 | 61.61 |
| PROTO-CLIP-$F$ | 3xConv | ✗ | 34.13 | 42.83 | **51.91** | **46.87** | 62.35 |
| PROTO-CLIP-$F$ | 3xConv | ✓ | **35.22** | **44.09** | 50.39 | 46.57 | 60.39 |

Table 4: Backbone ablation study. Dataset=FEWSOL-52 (P et al., 2023). $K = 16$.

**Shots.** Table 5 displays the results of using different numbers of shots on ImageNet (Deng et al., 2009) and FEWSOL (P et al., 2023). With more shots for training, the classification accuracy is improved accordingly. The choice of K=16 for our experiments aligns with the prevalent practice in the field of vision-language few-shot learning. This specific value has been widely adopted, as evidenced in various scholarly works such as (Gao et al., 2021; Zhou et al., 2022; Zhang et al., 2022) Moreover, given our specific emphasis on the few-shot context, it appeared prudent to exercise caution when surpassing a particular threshold, specifically 16 in our case.

As a result, we embarked on an ablation study involving the ImageNet (Deng et al., 2009) dataset. This particular dataset holds the largest number of classes (1000) and thus provided a suitable platform for investigating shots values beyond 16, such as 32 and 64. Despite our intention to explore 128 shots, our experimental hardware's memory limitations prohibited us from pursuing this avenue. Additionally, FEWSOL is valuable for few-shot object learning, especially in robotics. We capped shots at 16 for FEWSOL as average number of samples per class in FEWSOL hovers around 15. Consequently, we conjectured that going beyond might yield diminishing learning returns. These insights are detailed in Table 5.

| Dataset | Method | 1 | 2 | 4 | 8 | 16 | 32 | 64 |
|---|---|---|---|---|---|---|---|---|
| ImageNet (Deng et al., 2009) | Tip (Zhang et al., 2022) | **60.70** | **60.96** | 60.98 | 61.45 | 62.01 | 62.51 | 62.88 |
| | PROTO-CLIP | 60.31 | 60.64 | **61.30** | **62.12** | **62.77** | **62.98** | **63.23** |
| | Tip-F (Zhang et al., 2022) | **61.13** | **61.69** | **62.52** | 64.00 | 65.51 | 66.58 | **67.96** |
| | PROTO-CLIP-$F$ | 60.32 | 60.64 | 61.30 | 63.92 | 65.75 | 66.47 | 65.36 |
| | PROTO-CLIP-$F$-$Q^T$ | 59.12 | 60.48 | 61.80 | **64.03** | 65.91 | **66.71** | 66.90 |
| FEWSOL-52 (P et al., 2023) | Tip (Zhang et al., 2022) | **27.30** | 26.22 | 28.70 | 29.22 | 28.87 | ✗ | ✗ |
| | PROTO-CLIP | 27.09 | **28.35** | **29.13** | **29.83** | **29.96** | ✗ | ✗ |
| | Tip-F (Zhang et al., 2022) | **27.91** | **27.43** | 29.13 | 32.43 | 34.04 | ✗ | ✗ |
| | PROTO-CLIP-$F$ | 22.22 | 26.17 | 27.09 | **33.26** | **35.22** | ✗ | ✗ |
| | PROTO-CLIP-$F$-$Q^T$ | 21.65 | 25.91 | **30.30** | 32.70 | 34.70 | ✗ | ✗ |

Table 5: Shots ablation results. Backbone='CLIP ResNet50'. Since average number of samples in training classes are $\approx 15$ in FEWSOL-52, $K > 16$ are not considered for FEWSOL-52.

## 4.2 COMPARISON WITH OTHER METHODS

Table 6 shows the performance of PROTO-CLIP compared to the state-of-the-art methods using CLIP for few-shot learning in the literature: Linear-Probe CLIP (Radford et al., 2021), CoOp (Zhou et al., 2022), CLIP-Adapter (Gao et al., 2021) and Tip-Adapter (Zhou et al., 2022). We follow these methods and use CLIP's ResNet50 backbone for this comparison. The fine-tuned variant of Tip-Adapter "Tip-F" is the most competitive method compared to ours. The performance of PROTO-CLIP on very few shots, i.e., 1 shot and 2 shots is inferior compared to Tip-F. When the number of shots increases to 4, 8 and 16, the fine-tuned variants of PROTO-CLIP outperform Tip-F. The enhanced performance of our proposed PROTO-CLIP method can be attributed to its reliance on robust image and textual prototypes, which subsequently leads to improved classification accuracy. Therefore, our model benefits from more than 4 shots, while it is not as good as Tip-F when using 1 shot and 2 shots. PROTO-CLIP-$F$-$Q^T$ performs better than PROTO-CLIP-$F$ on most datasets by using the data augmentation of query images during training. Please see the appendix for more details.

| Dataset # classes | ImageNet 1000 | FGVC 100 | Pets 37 | Cars 196 | EuroSAT 10 | Caltech101 100 | SUN397 397 | DTD 47 | Flowers 102 | Food101 101 | UCF101 101 | FEWSOL 52 |
|---|---|---|---|---|---|---|---|---|---|---|---|---|
| Zero-shot CLIP (Radford et al., 2021) | 60.33 | 17.10 | 85.83 | 55.74 | 37.52 | 85.92 | 58.52 | 42.20 | 66.02 | 77.32 | 61.35 | 25.91 |
| **1 shot** | | | | | | | | | | | | |
| Linear-Probe CLIP (Radford et al., 2021) | 22.07 | 12.89 | 30.14 | 24.64 | 51.00 | 70.62 | 32.80 | 29.59 | 58.07 | 30.13 | 41.43 | - |
| CoOp (Zhou et al., 2022) | 57.15 | 9.64 | 85.89 | 55.59 | 50.63 | 87.53 | 60.29 | 44.39 | 68.12 | 74.32 | 61.92 | - |
| CLIP-A (Gao et al., 2021) | 61.20 | 17.49 | 85.99 | 55.13 | 61.40 | 88.60 | 61.30 | 45.80 | 73.49 | 76.82 | 62.20 | - |
| Tip (Zhang et al., 2022) | 60.70 | 19.05 | 86.10 | 57.54 | 54.38 | 87.18 | 61.30 | 46.22 | 73.12 | 77.42 | 62.60 | 27.30 |
| Tip-F (Zhang et al., 2022) | **61.13** | **20.22** | **87.00** | 58.86 | 59.53 | **89.33** | **62.50** | **49.65** | **79.98** | 77.51 | 64.87 | **27.91** |
| PROTO-CLIP | 60.31 | 19.59 | 86.10 | 57.29 | 55.53 | 87.99 | 60.81 | 46.04 | 76.98 | 77.36 | 63.15 | 27.09 |
| PROTO-CLIP-$F$ | 60.32 | 19.50 | 85.72 | 57.34 | 54.93 | 88.07 | 60.83 | 45.86 | 77.34 | 77.34 | 63.07 | 22.22 |
| PROTO-CLIP-$F$-$Q^T$ | 59.12 | 16.26 | 83.62 | 52.77 | **61.95** | 88.48 | 61.43 | 32.27 | 68.53 | 75.16 | 62.44 | 21.65 |
| **2 shots** | | | | | | | | | | | | |
| Linear-Probe CLIP (Radford et al., 2021) | 31.95 | 17.85 | 43.47 | 36.53 | 61.58 | 78.72 | 44.44 | 39.48 | 73.35 | 42.79 | 53.55 | - |
| CoOp (Zhou et al., 2022) | 57.81 | 18.68 | 82.64 | 58.28 | 61.50 | 87.93 | 59.48 | 45.15 | 77.51 | 72.49 | 64.09 | - |
| CLIP-A (Gao et al., 2021) | 61.52 | 20.10 | 86.73 | 58.74 | 63.90 | 89.37 | 63.29 | 51.48 | 81.61 | 77.22 | 67.12 | - |
| Tip (Zhang et al., 2022) | 60.96 | 21.21 | 87.03 | 57.93 | 61.68 | 88.44 | 62.70 | 49.47 | 79.13 | 77.52 | 64.74 | 26.22 |
| Tip-F (Zhang et al., 2022) | **61.69** | **23.19** | 87.03 | **61.50** | 66.15 | **89.74** | 63.64 | 53.72 | 82.30 | **77.81** | 66.43 | 27.43 |
| PROTO-CLIP | 60.64 | 22.14 | **87.38** | 60.01 | 64.89 | 89.05 | 63.12 | 51.06 | 83.39 | 77.34 | 67.46 | **28.35** |
| PROTO-CLIP-$F$ | 60.64 | 22.14 | **87.38** | 60.04 | 64.86 | 89.09 | 63.20 | 51.85 | **83.52** | 77.34 | 67.49 | 26.17 |
| PROTO-CLIP-$F$-$Q^T$ | 60.48 | 20.01 | 85.28 | 60.02 | 63.59 | 89.49 | **65.46** | 45.69 | 81.20 | 76.15 | **68.83** | 25.91 |
| **4 shots** | | | | | | | | | | | | |
| Linear-Probe CLIP (Radford et al., 2021) | 41.29 | 23.57 | 56.35 | 48.42 | 68.27 | 84.34 | 54.59 | 50.06 | 84.80 | 55.15 | 62.23 | - |
| CoOp (Zhou et al., 2022) | 59.99 | 21.87 | 86.70 | 62.62 | 70.18 | 89.55 | 63.47 | 53.49 | 86.20 | 73.33 | 67.03 | - |
| CLIP-A (Gao et al., 2021) | 61.84 | 22.59 | 87.46 | 62.45 | 73.38 | 89.98 | 65.96 | 56.86 | 87.17 | 77.92 | 69.05 | - |
| Tip (Zhang et al., 2022) | 60.98 | 22.41 | 86.45 | 61.45 | 65.32 | 89.39 | 64.15 | 53.96 | 83.80 | 77.54 | 66.46 | 28.70 |
| Tip-F (Zhang et al., 2022) | **62.52** | 25.80 | **87.54** | 64.57 | 74.12 | 90.56 | 66.21 | **57.39** | 88.83 | **78.24** | 70.55 | 29.13 |
| PROTO-CLIP | 61.30 | 23.25 | 87.19 | 63.33 | 68.67 | 89.57 | 65.51 | 55.91 | 88.23 | 77.58 | 69.50 | 29.13 |
| PROTO-CLIP-$F$ | 61.30 | 23.31 | 86.95 | 63.34 | 68.52 | 89.62 | 65.57 | 57.21 | 88.27 | 77.58 | 69.55 | 27.09 |
| PROTO-CLIP-$F$-$Q^T$ | 61.80 | **27.63** | 87.11 | **66.24** | **80.64** | **91.81** | 68.09 | 56.86 | **89.85** | 76.94 | 70.16 | **30.30** |
| **8 shots** | | | | | | | | | | | | |
| Linear-Probe CLIP (Radford et al., 2021) | 49.55 | 29.55 | 65.94 | 60.82 | 76.93 | 87.78 | 62.17 | 56.56 | 92.00 | 63.82 | 69.64 | - |
| CoOp (Zhou et al., 2022) | 61.56 | 26.13 | 85.32 | 68.43 | 76.73 | 90.21 | 65.52 | 59.97 | 91.18 | 71.82 | 71.94 | - |
| CLIP-A (Gao et al., 2021) | 62.68 | 26.25 | 87.65 | 67.89 | 77.93 | 91.40 | 67.50 | 61.00 | 91.72 | 78.04 | 73.30 | - |
| Tip (Zhang et al., 2022) | 61.45 | 25.59 | 87.03 | 62.93 | 67.95 | 89.83 | 65.62 | 58.63 | 87.98 | 77.76 | 68.68 | 29.22 |
| Tip-F (Zhang et al., 2022) | 64.00 | 30.21 | 88.09 | 69.25 | 77.93 | 91.44 | 68.87 | 62.71 | 91.51 | **78.64** | 74.25 | 32.43 |
| PROTO-CLIP | 62.12 | 27.63 | 88.04 | 64.93 | 69.42 | 90.22 | 67.37 | 59.34 | 92.08 | 77.90 | 71.08 | 29.83 |
| PROTO-CLIP-$F$ | 63.92 | 31.32 | **88.55** | 70.35 | 78.94 | 92.54 | 69.59 | 62.35 | 93.79 | 78.29 | 74.81 | **33.26** |
| PROTO-CLIP-$F$-$Q^T$ | **64.03** | 35.82 | 87.46 | 71.50 | 81.89 | 92.62 | 70.02 | 64.01 | 94.28 | 78.61 | 75.34 | 32.70 |
| **16 shots** | | | | | | | | | | | | |
| Linear-Probe CLIP (Radford et al., 2021) | 55.87 | 36.39 | 76.42 | 70.08 | 82.76 | 90.63 | 67.15 | 63.97 | 94.95 | 70.17 | 73.72 | - |
| CoOp (Zhou et al., 2022) | 62.95 | 31.26 | 87.01 | 73.36 | 83.53 | 91.83 | 69.26 | 63.58 | 94.51 | 74.67 | 75.71 | - |
| CLIP-A (Gao et al., 2021) | 63.59 | 32.10 | 87.84 | 74.01 | 84.43 | 92.49 | 69.55 | 65.96 | 93.90 | 78.25 | 76.76 | - |
| Tip (Zhang et al., 2022) | 62.02 | 29.76 | 88.14 | 66.77 | 70.54 | 90.18 | 66.85 | 60.93 | 89.89 | 77.83 | 70.58 | 28.87 |
| Tip-F (Zhang et al., 2022) | 65.51 | 35.55 | **89.70** | 75.74 | 84.54 | 92.86 | 71.47 | 66.55 | 94.80 | **79.43** | 78.03 | 34.04 |
| PROTO-CLIP | 62.77 | 29.67 | 88.61 | 68.11 | 72.95 | 91.08 | 68.09 | 61.64 | 92.94 | 78.11 | 73.35 | 29.96 |
| PROTO-CLIP-$F$ | 65.75 | 37.56 | 89.62 | 75.25 | 83.53 | 93.43 | **71.94** | **68.56** | 95.78 | 79.09 | 77.50 | **35.22** |
| PROTO-CLIP-$F$-$Q^T$ | **65.91** | **40.65** | 89.34 | **76.76** | **86.59** | **93.59** | 72.19 | 68.50 | **96.35** | 79.34 | **78.11** | 34.70 |

Table 6: Few-shot classification results of various CLIP based few shot learning methods on different datasets across various shots using the CLIP ResNet50 backbone.

### 4.3 REAL WORLD EXPERIMENTS

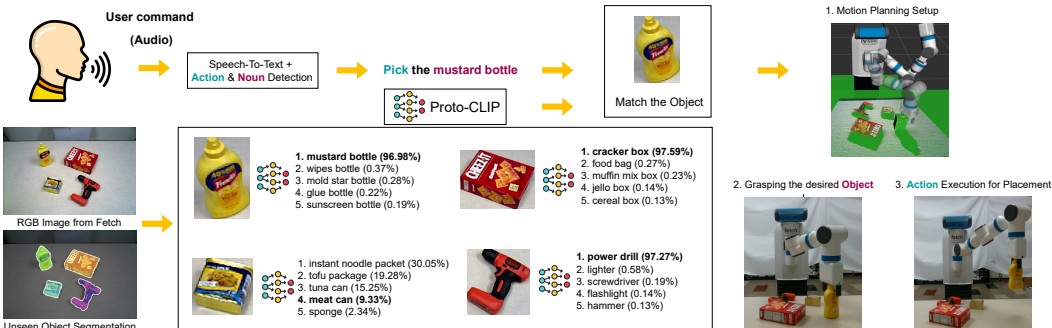

Figure 5: Results for the real world setup with top-5 predictions from the PROTO-CLIP-$F$ (ViT-L/14) model trained on FEWSOL-198 (P et al., 2023). The Speech-To-Text is performed via Whisper (Radford et al., 2022).

As an application, we have built a robotic system to verify the effectiveness of PROTO-CLIP for object recognition in the real world. Fig. 5 illustrates our pipeline for the system. It takes human instruction in the form of voice commands as input such as "pick something" or "grasp something". The system first applies Automatic Speech Recognition (ASR) to convert voice input to text using OpenAI Whisper (Radford et al., 2022). Then the system grounds the noun in the human instruction into a target object observed from an input image. This is achieved by joint object segmentation and classification. We utilize unseen object instance segmentation (Lu et al., 2022) to segment objects in cluttered scenes and then classify each segmented object with PROTO-CLIP. By matching the noun with the class labels, the system can ground the target in the image. Once the target object is recognized, we use Contact-GraspNet (Sundermeyer et al., 2021) for grasp planning and MoveIt motion planning toolbox (Chitta et al., 2012) to pick and place the target. Please see the supplementary material for more real-world results.

## 5 LIMITATIONS

PROTO-CLIP performs poorly in low-shot regimes, as is evident from Table 6. A hyperparameter grid search is necessary for each new dataset, following the methodology of Tip-Adapter. This requirement applies to every combination of the new dataset and the backbone. Embracing the diversity of datasets, our system thrives on the need for different set-ups. When encountering a new dataset, we actively compare the effectiveness of $F$ and $F$-$Q^T$ to determine the optimal choice. This dynamic approach transforms the potential weakness into a strength, allowing us to adapt and maximize performance for every unique dataset. During our observations, we discovered that data transformations play a crucial role in building the memory model.

## 6 CONCLUSION AND FUTURE WORK

We have introduced a novel method for few-shot learning based on the CLIP (Radford et al., 2021) vision-language model. Our method learns image prototypes and text prototypes from few-shot training examples and aligns the corresponding image-text prototypes for classification. The model is equipped with learnable image memory and text memory for support images and a learnable adapter for query images. Compared to previous CLIP-based few-shot learning methods, our method is flexible in configuring these learnable components, resulting in powerful learned models.

Good feature representation is the key in few-shot learning. Future work includes how to further improve feature representation learning compared to CLIP models. One idea is to adapt more powerful vision-language models such as GPT variants. The FEWSOL (P et al., 2023) dataset also provides multiview and depth information about objects. Exploring this 3D information in few-shot object recognition is also a promising direction.

REPRODUCIBILITY STATEMENT

In accordance with the commitment to transparency and reproducibility, we provide the following information to ensure the replicability of our research:

**Code Availability:** We are committed to making our code available upon acceptance. Currently, it is not in an anonymized state, but we will swiftly share it to facilitate replication of our experiments.

**Python Package Toolkit:** To enable the execution of real-world robot experiments as described in our paper, we will provide a dedicated Python package toolkit. This toolkit will include both the original code and additional resources necessary for conducting the experiments. Please see the supplementary videos showcasing our real-world experiments on a Fetch Robot.

**Random Number Seeds:** We have employed seeds for random number generation throughout our experiments to ensure reproducibility. These seeds will be clearly documented and shared to facilitate the recreation of our results.

**Dataset Splits:** For the experiments involving FEWSOL P et al. (2023), we have supplied a splits file as for all the other 11 datasets splits files were available as shared by Tip-Adapter (Zhang et al., 2022). We emphasize the importance of using this splits file to maintain the principles of reproducibility and to ensure a fair and transparent comparison with prior methodologies. Further details can be found in the appendix.

We believe that these measures, along with the comprehensive information provided in the main paper, appendix, and supplementary materials, will empower researchers to effectively replicate and build upon our work. Our commitment to reproducibility aligns with the standards of ICLR, and we are dedicated to supporting the scientific community in advancing the field.

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

APPENDIX

# A EXPERIMENTS

## A.1 $NK$ SETUP

Our research focuses on addressing the $N$-way $K$-shot classification problem, which involves the classification of data into $N$ classes using $K$ samples per class. In our approach, we organize the problem into episodes, with each episode comprising a set of $N$ classes and $K$ samples from each class. In contrast to the methodology employed in Tip-Adapter (Zhang et al., 2022), where all classes are considered for episode composition, we adopt a random sampling technique for the selection of classes to form the episodes. This sampling protocol ensures that all the classes are encountered within a single epoch of the training process.

By incorporating random sampling of classes, we hope to enhance the diversity and generalizability of the training episodes. This technique ensures that the model encounters a wide range of classes during the training phase, facilitating better learning and improved performance on unseen data. By adopting a sophisticated sampling protocol and incorporating random class selection, our method demonstrates a robust and comprehensive approach to the $N$-way $K$-shot classification problem. Using diverse training episodes enhances the model's ability to generalize to new classes and improves its overall performance on various classification tasks.

## A.2 DATASET SPLITS FILE

In accordance with the established practices set forth by Tip-Adapter (Zhang et al., 2022), we adopted the identical splits file for our experimental analysis, encompassing a total of 11 datasets. Since, we want to improve object classification for robot manipulation tasks, FEWSOL (P et al., 2023) is one additional dataset that we have added to the pool of 11 datasets considered in Tip-Adapter (Zhang et al., 2022) which is highly beneficial for few shot learning in robotic environments. This approach ensures a fair and meaningful comparison with the findings presented in Tip-Adapter (Zhang et al., 2022) and FEWSOL (P et al., 2023), where an extensive investigation into the efficacy of CLIP (Radford et al., 2021) and its variants has been meticulously demonstrated.

A splits file serves as a crucial component, providing comprehensive information regarding the train/val/test split for each dataset. This information is instrumental in facilitating the training, hyperparameter search and evaluation processes. We have created new splits file for 4 variants as in FEWSOL (P et al., 2023): (i) FEWSOL-11 (ii) FEWSOL-41 (iii) FEWSOL-52 (iv) FEWSOL-198. The $C$ in FEWSOL-$C$ indicates the number of classes in the splits.

By adhering to the utilization of the same splits file, we strive to uphold the principles of reproducibility and promote a transparent and equitable comparison with previous methodologies.

We highly encourage the adoption of a similar mechanism in future research endeavors within this domain. Such adherence to consistent practices would not only contribute to the reproducibility of results but also foster a fair and valid comparison to previous methods. By aligning our methodologies, the scientific community can effectively build upon existing knowledge and drive the field forward in a robust and meaningful manner.

| FEWSOL-C | Description |
|---|---|
| FEWSOL-11 | Consists of 11 classes overlapping between synthetic 125 classes and real 198 classes |
| FEWSOL-41 | Consists of 41 classes which are not present in synthetic classes |
| FEWSOL-52 | Consists of union of FEWSOL-41 and FEWSOL-11 |
| FEWSOL-198 | Consists of all of the real classes with real world testing images as mentioned in Section. D |

Table 7: Description of 4 variants of FEWSOL (P et al., 2023) dataset. Splits file of each are shared with this work. We have considered FEWSOL-52 (P et al., 2023) and FEWSOL-198 (P et al., 2023) in our experiments.

## A.3 HYPERPARAMETERS

| Hyperparameter | Description |
|---|---|
| $\alpha$ | Controls the contribution from image and text memory bank |
| $\beta$ | Controls the sharpness of the logits. Acts like a temperature parameter as in (Hinton et al., 2015) |
| $K$ | Shots to be considered for building the visual and textual memory bank |
| Train-Text-Memory | A boolean flag indicating whether to train text memory bank |
| Adapter | The alias of the available adapters in PROTO-CLIP: ['3xConv', '2xConv', 'MLP'] |
| Backbone | The alias of the available CLIP (Radford et al., 2021) backbones: ['RN50', 'RN101', 'ViT-B/16', 'ViT-B/32', 'ViT-L/14'] |

Table 8: Description of hyperparameters considered in PROTO-CLIP experiments.

The table provides a comprehensive overview of the hyperparameters considered in the PROTO-CLIP experiments. These hyperparameters play a crucial role in configuring and fine-tuning the model's behavior, allowing researchers to explore different settings and optimize its performance based on specific requirements and objectives.

$\alpha$. This hyperparameter controls the contribution from the image and text memory bank. It determines the balance between visual and textual information during the classification process. By adjusting the value of $\alpha$, researchers can emphasize the importance of either modality, enabling the model to effectively leverage the strengths of both image and text representations.

$\beta$. Acting as a sharpness control parameter, $\beta$ influences the logits' sharpness in the classification process. It functions similarly to a temperature parameter, as observed in the work of (Hinton et al., 2015). By manipulating $\beta$, researchers can control the spread of probabilities assigned to different classes, thus affecting the model's confidence levels in its predictions.

**Adapter.** The adapter hyperparameter refers to the alias of the available adapters in PROTO-CLIP. In the context of PROTO-CLIP, the framework offers a selection of adapters, each associated with a specific alias. The available aliases for the adapters are as follows: ['3xConv', '2xConv', 'MLP']. In the main paper, these aliases are further clarified and correspond to the following adapter configurations:

- 3xConv: This alias represents an adapter configuration consisting of three convolutional layers. Each layer applies a set of filters to the input data, enabling the model to extract and learn relevant features through hierarchical transformations. The '3xConv' adapter offers increased expressive power and potential for capturing intricate patterns within the data.

- 2xConv: This alias denotes an adapter configuration comprising two convolutional layers. Similar to the '3xConv' adapter, the '2xConv' adapter utilizes convolutional operations to extract meaningful features from the input data. Although it has a slightly simpler architecture than the '3xConv' adapter, the '2xConv' adapter still maintains a notable capacity for capturing and representing essential characteristics of the data.

- MLP: The 'MLP' alias refers to the adapter configuration known as MLP, which stands for Multi-Layer Perceptron. The MLP adapter consists of one hidden layer with a size equal to $D/4$, where $D$ represents the embedding size derived from the CLIP backbone. The MLP adapter employs fully connected layers, allowing for non-linear transformations and the integration of complex interactions between the input features.

By leveraging these different adapter configurations, researchers can tailor the behavior of PROTO-CLIP to their specific needs and experiment with various levels of model complexity. Each adapter variant offers distinct architectural characteristics and capabilities, enabling the model to capture and utilize different levels of abstraction and contextual information from the input data.

**K**. This hyperparameter determines the number of shots considered for building the visual and textual memory bank. Shots refer to the number of examples available per class in the training set. By adjusting the value of $K$, researchers can control the amount of information captured in the memory bank, potentially influencing the model's ability to generalize and recognize novel instances of the classes.

By carefully selecting and adjusting these hyperparameters, researchers can tailor the behavior and performance of the PROTO-CLIP model to suit their specific research goals. Fine-tuning these parameters empowers researchers to explore different trade-offs and discover optimal configurations that maximize classification accuracy, generalization capabilities, and efficiency.

## B  EXTENDED ABLATION STUDY

### B.1  OUT OF DISTRIBUTION EXPERIMENTS

In our evaluation, we assessed the out-of-distribution capabilities of our proposed PROTO-CLIP models by training them on one and then testing on two different datasets. Specifically, we used ImageNet (Deng et al., 2009) as the source dataset, providing a 16-shot training set, and conducted testing on ImageNetV2 (Recht et al., 2019) and ImageNet-Sketch (Wang et al., 2019), which contain categories similar to ImageNet but with some semantic differences.

The results, as presented in Table 9, demonstrate that PROTO-CLIP exhibits remarkable robustness when faced with distribution shifts. It outperforms the baselines on ImageNet-V2 and is on par on ImageNet-Sketch, highlighting the advantages of the alignment in out-of-distribution evaluation.

| Datasets | Source | Target | |
|---|---|---|---|
| | **ImageNet** | **-V2** (Recht et al., 2019) | **-Sketch** (Wang et al., 2019) |
| Zero-Shot-CLIP (Radford et al., 2021) | 60.33 | 53.27 | 35.44 |
| Linear Probe CLIP (Radford et al., 2021) | 56.13 | 45.61 | 19.13 |
| CoOp (Zhou et al., 2022) | 62.95 | 54.58 | 31.04 |
| CLIP-Adapter (Gao et al., 2021) | 63.59 | 55.69 | 35.68 |
| Tip (Zhang et al., 2022) | 62.03 | 54.60 | 35.90 |
| Tip-F (Zhang et al., 2022) | 65.51 | 57.11 | **36.00** |
| PROTO-CLIP | 62.77 | 55.23 | 35.62 |
| PROTO-CLIP-$F$ | 65.75 | 56.84 | 35.29 |
| PROTO-CLIP-$F$-$Q^T$ | **65.91** | **57.32** | 35.99 |

Table 9: Out of distribution accuracy study using Imagenet-V2 (Recht et al., 2019) and ImageNet-Sketch (Wang et al., 2019) datasets.

### B.2  ADAPTER AND TEXT MEMORY BANK ABLATION ON FEWSOL-198 DATASET

| Adapter | Train-Text-Memory | Top-1 Accuracy | $\Psi$ |
|---|---|---|---|
| MLP | ✗ | **68.75** | **6** |
| MLP | ✓ | 68.75 | 3 |
| 2xConv | ✗ | 65.62 | 12 |
| 2xConv | ✓ | 62.50 | 57 |
| 3xConv | ✗ | 65.62 | 3 |
| 3xConv | ✓ | 68.75 | 1 |

Table 10: Adapter ablation study. Model=PROTO-CLIP-$F$. Dataset=FEWSOL-198 (P et al., 2023). $K = 16$. $\Psi$ is the number of $\alpha, \beta$ combinations for which the max accuracy was obtained. Based on $\Psi$, MLP without training the text memory is the best configuration for FEWSOL P et al. (2023). Out of the 6 $\alpha, \beta$ combinations, we selected one randomly.

## C  T-SNE PLOT

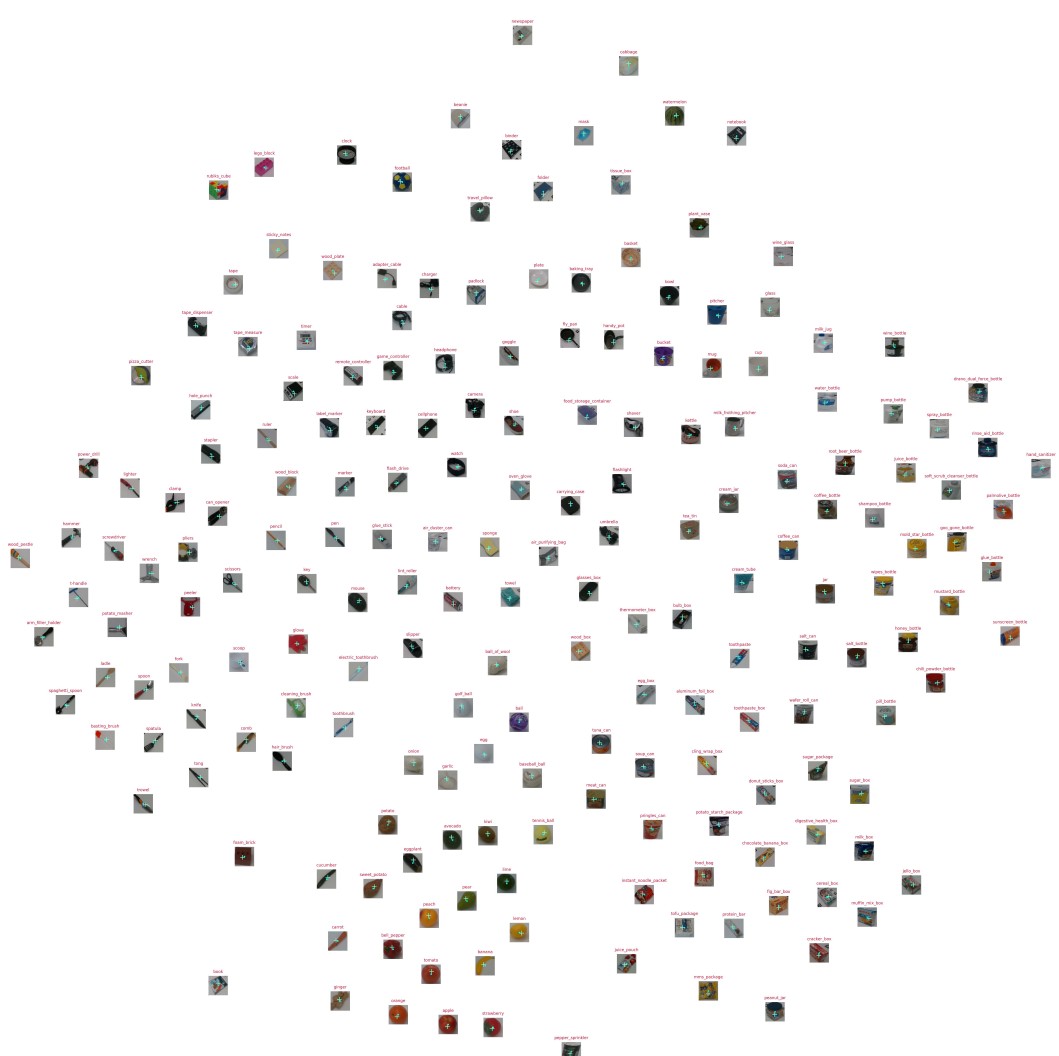

Figure 6: t-SNE plot after training PROTO-CLIP-$F$ on FEWSOL-198 (P et al., 2023) using CLIP ViT-L/14 backbone, where · and + indicate the image and text prototypes of the class displayed by the object image respectively.

## D  REAL-WORLD-TESTING AS IN FEWSOL (P ET AL., 2023)

Following FEWSOL (P et al., 2023), in this experiment, we aim to build a few-shot classification model that works best on real-world perception systems. We train PROTO-CLIP with all real data from the FEWSOL dataset, i.e 198 classes and then test the trained model in our lab on the task of joint object segmentation and few-shot classification experiment. The pipeline consists of (i) Collecting RGB-D images from a Fetch mobile manipulator (ii) Unseen object segmentation using UCN (Xiang et al., 2021b) and (iii) Few shot classification using PROTO-CLIP. We tested on 32 objects with 4 objects in an image scene.

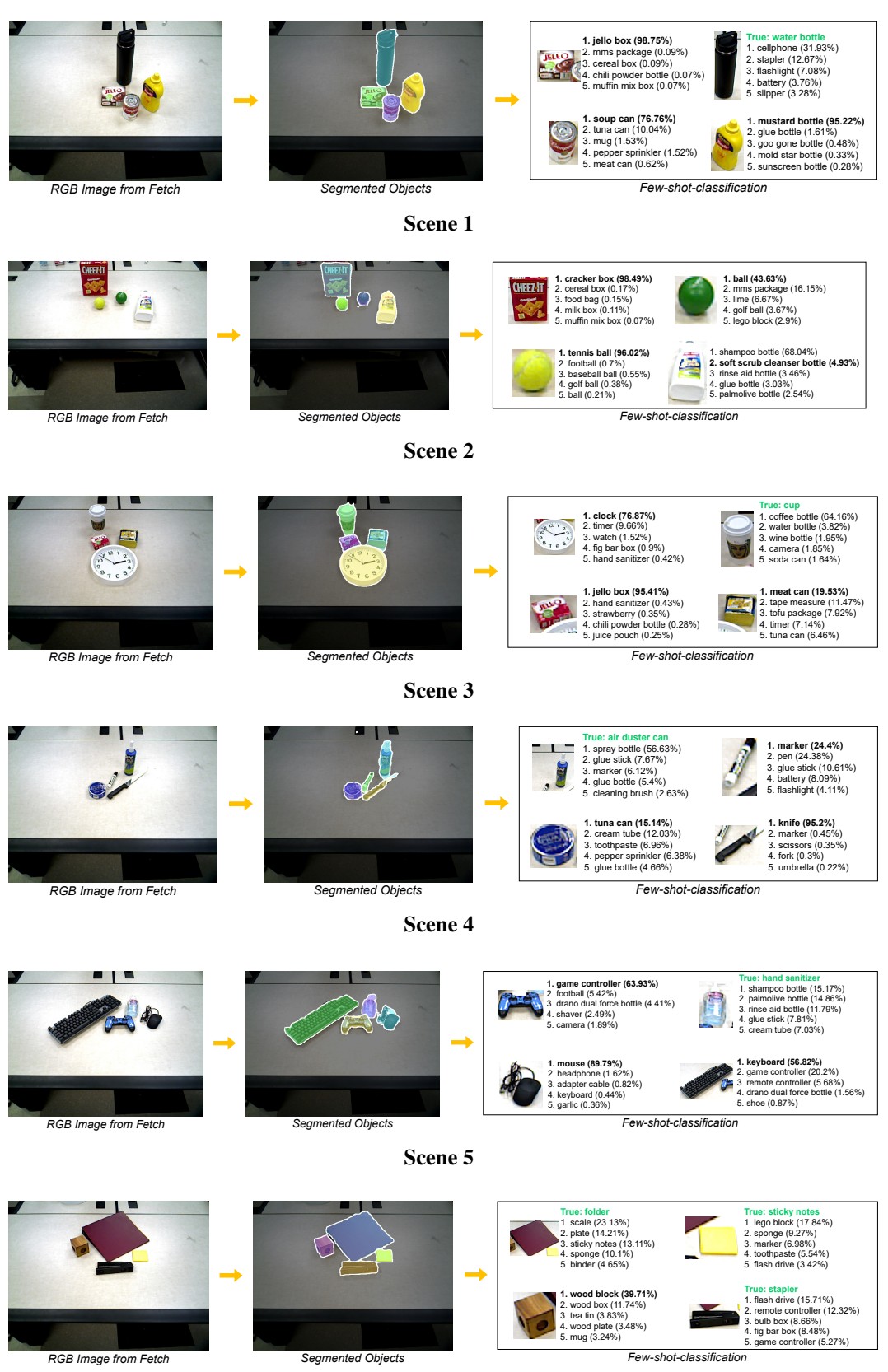

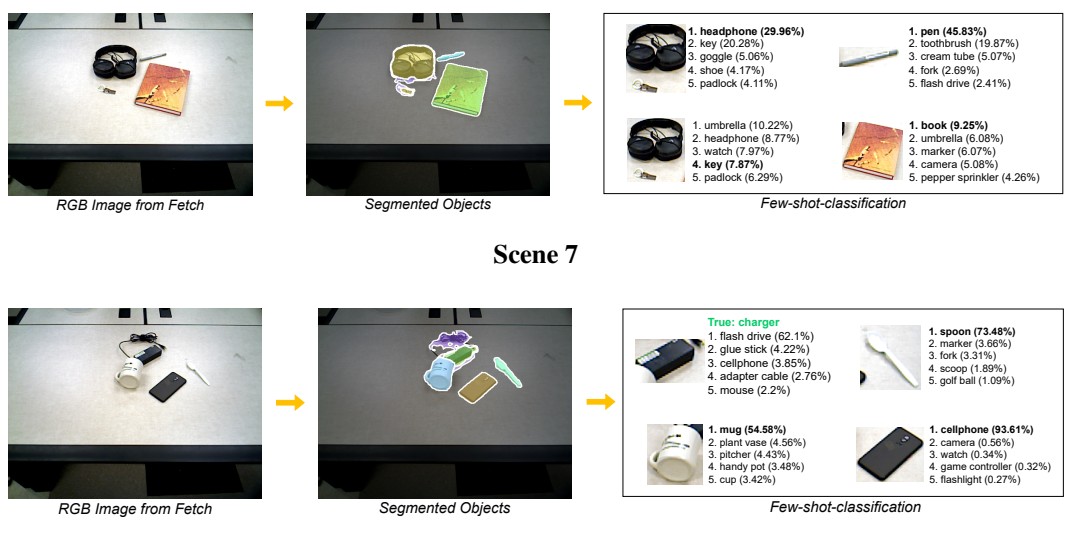

Figure 6: Top-5 predictions of our PROTO-CLIP-*F* model trained on FEWSOL-198 for 8 real world scenes as used in FEWSOL (P et al., 2023).

# E  FREQUENTLY ASKED QUESTIONS (FAQ)

**Q: How many text prompts have been used in the experiments?**

**A:** All datasets except ImageNet (Deng et al., 2009) have a single text prompt template. ImageNet uses 7. For comparison purpose, this setting has been borrowed from Tip-Adapter (Zhang et al., 2022). More text prompts can be used.

**Q: Why has FEWSOL been used for real world experiments?**

**A:** For a robot to work in human environments like kitchen, living room etc., it has to interact with various daily objects. FEWSOL (P et al., 2023) comes in handy when thinking of learning good representations of daily objects for manipulation tasks. Hence, we chose to experiments with FEWSOL.

**Q: Describe the real-world experiments and its outcomes?**

**A:** Following the approach taken in FEWSOL (P et al., 2023), we have conducted a comprehensive study involving 32 real-world objects. Our method achieved top-1 accuracy of 68.75%, while Tip-Adapter demonstrates a top-1 accuracy of 65.63%. The model used was trained on FEWSOL-198. For a detailed analysis, kindly refer to the appendix section D. Since the experiments have an external dependency on a custom segmentation method, the quality of segmentation also affects the classification performance. Although we conjecture that more classes could make the classification problem difficult, the current performance of PROTO-CLIP makes room for healthy classification and future research. Moreover, we have also executed four real-world experiments focusing on user-command-oriented grasping. These experiments utilized PROTO-CLIP predictions and involved four objects each time, with a total of 16 objects placed on a tabletop. The supplementary videos have been updated with video evidence showcasing the efficacy of our approach. We invite you to review these materials at your convenience. Few key classifications during real world experiments:

- **Instance level:** Different colored bell peppers identified correctly
- **Fine grained:** E.g. Lime vs lemon identified correctly
- **Clutter:** E.g. Scissors on top of umbrella identified correctly

**Q: Any specific observations during inference?**

**A:** Object segmentation and orientation matters. Segmentation is more important as in clutter scenes a bad segmentation can cause problems in classification. Moreover, lighting conditions also play a key role as it impacts classification of shiny objects.

**Q: What are some potential explanations for the difference in behavior between Tip-Adapter and PROTO-CLIP?**

**A:** We think that the behaviour difference is due to the mechanisms that are employed in each settings. Tip-Adapter works by computing support query affinity and then comparing the closeness to the possible classes in textual form which in turn helps in generating the probability distribution for classification. Proto-CLIP on the other hand utilizes the prototypes build from the visual and textual memory banks learned during few shot training and then classifies the incoming query image based on the probability distribution created by the contribution of visual and textual prototypes w.r.t. to the given query image.

**Q: Why the proposed method perform better when K=4, 8, or 16?**

**A:** The enhanced performance of our proposed PROTO-CLIP method can be attributed to its reliance on robust image and textual prototypes, which subsequently leads to improved classification accuracy. In our approach, each embedding within the visual memory bank is computed through multiple augmentations (typically around 10) of a specific image sample. While we did indeed explore the use of augmentations, we found that the inclusion of a greater number of sample images yielded superior outcomes. We hold the perspective that a larger quantity of high-quality samples introduces a richer array of information encompassing texture, lighting, orientation, color, and shapes. This wealth of information significantly contributes to the establishment of more resilient prototypes, thereby fortifying the entire classification process.

**Q: How the proposed and baseline methods perform differently across different datasets?**

**A:** As evident from the results in Table 6, in extremely low shot scenarios, e.g. $K = 1$, PROTO-CLIP variants show competing results w.r.t. the baseline Tip-Adapter (Zhang et al., 2022). As K increases, PROTO-CLIP starts to show promising results by outperforming the baseline in more datasets.

- $K = 2$, outperforms on $5/12$.
- $K = 4$, outperforms on $7/12$.
- $K = 8$, outperforms on $11/12$ and on par with the remaining one dataset.
- $K = 16$, outperforms on $10/12$ and on par with the remaining two datasets; which shows that more shots might not be good for some datasets.

Moreover, Table 2 and 3 show the effect of different query adapters and loss functions across datasets. Hence, it wouldn't hurt to say that different datasets have different needs which can be attributed to its properties.

**Q: What is the requirement of PROTO-CLIP when we have SAM (Kirillov et al., 2023a)?**

**A:** SAM (Kirillov et al., 2023a) presents a promising approach, although it is not without its limitations. It necessitates the provision of prompts (e.g., Points, Bounding Boxes, Text), or alternatively, the segmentation of all elements within an image. In the former scenario, utilizing any of the prompts aside from textual cues requires the incorporation of heuristics or well-trained models to get reasonable point or bounding boxes to avoid inadvertent area mask predictions. Addressing how to accomplish this in the absence of human input is a distinct question. Regarding text prompts, the SAM paper itself acknowledges its status as a proof of concept. SAM relies on CLIP embeddings when employing text prompts, yet the provided sample results also fall short. To achieve reasonably accurate predictions, supplementary prompts like points or bounding boxes are employed alongside text prompts. In the latter case, where complete image segmentation is pursued, additional post-processing steps are essential to obtain pertinent masks from the generated outputs pertaining to over segmentation, under

segmentation and unwanted masks issue. Consequently, while SAM stands as an impressive model, its direct application to robotics use cases is not straightforward; it necessitates integration within a pipeline arrangement. In our endeavors, we incorporated SAM in a cluttered scene setup, yet the outcomes were less than satisfactory. Therefore, we propose that a lightweight model such as ours could potentially enhance SAM's predictions or synergize with analogous methodologies, ultimately bolstering downstream robotic tasks like manipulation or control. Please see the following Study on SAM (Kirillov et al., 2023b).

## F A STUDY ON SAM (KIRILLOV ET AL., 2023B)

Although the Segment Anything Model (SAM) (Kirillov et al., 2023b) is a great method for segmentation, we feel that it's too early to show good performance in robotics context. Clutter scene as shown in Fig. 7 is a prominent example of a robotics environment. A robot will encounter objects in a cluttered scene more often than clean ones. As we can see here, 2 runs of SAM on the same input image yield different results. There is over-segmentation as well as under-segmentation. Occlusion breaks the object connectivity and the masks generated are disjoint. There are several unwanted masks that need post-processing based on some heuristic in absence of a human. Here, we have used the segment entire image functionality. In order to use the prompt feature of SAM, we need a point location or bounding box as prompt which is possible in presence of a human or some mechanism that can yield good intended masks. In absence of a human, some estimation or approximation method needs to be used. Thus, directly applying SAM in robotics might not be a good option. However, it can act as a module in a large robotic pipeline for downstream tasks.

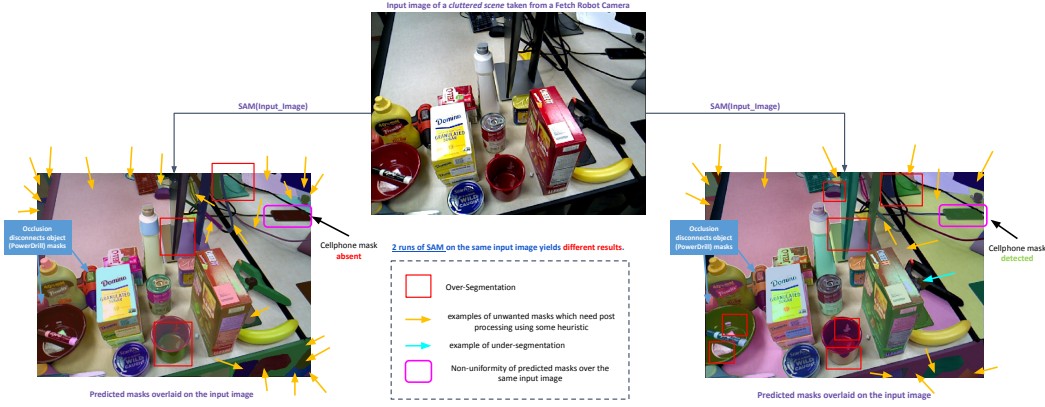

Figure 7: Execution of SAM (Kirillov et al., 2023b) on a sample real world clutter scene created in our lab.

Our PROTO-CLIP model can join forces with the predictions of SAM for better classification of the detected masks (after removing the unwanted masks). Unseen object segmentation methods like (Xie et al., 2021; Xiang et al., 2021a; Xie et al., 2020; Lu et al., 2023) are specifically targeted towards finding masks of **novel objects** in different scenes hence we believe that they have an upper hand in robotics context as they can aid robot manipulation tasks on novel objects. Fig. 7 (Scene:1-8) show the predictions of SAM (Kirillov et al., 2023b) vs UCN (Xiang et al., 2021a). PROTO-CLIP can be used in combination with any of these for downstream tasks involving vision-language modalities.

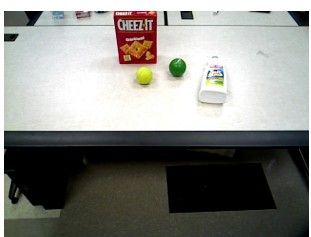 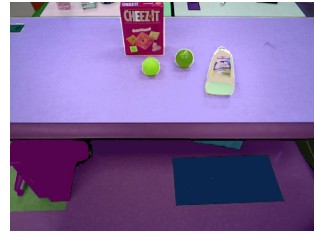 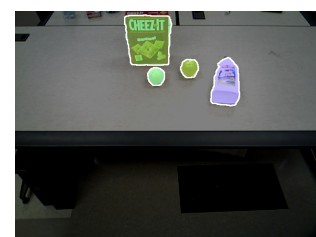

*RGB Image from Fetch*     *SAM Segmentation*     *UCN Segmentation*

**Scene-1**

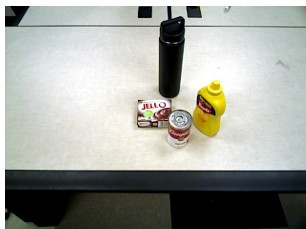 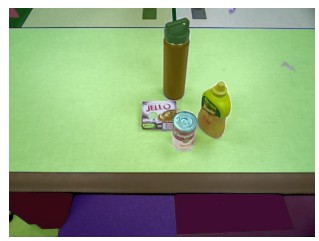 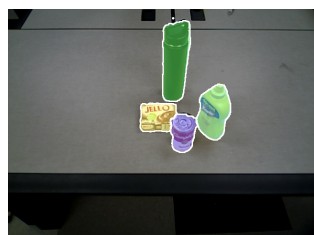

*RGB Image from Fetch*     *SAM Segmentation*     *UCN Segmentation*

**Scene-2**

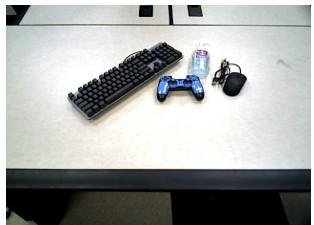 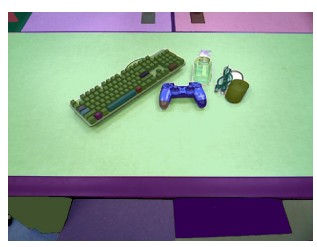 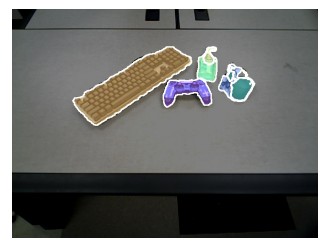

*RGB Image from Fetch*     *SAM Segmentation*     *UCN Segmentation*

**Scene-3**

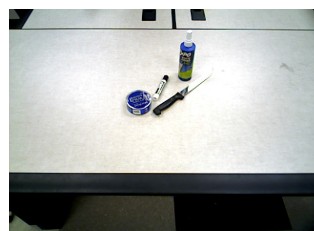 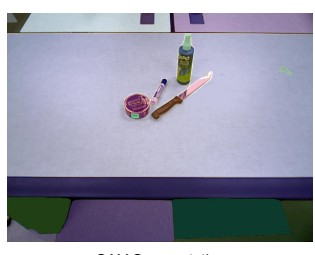 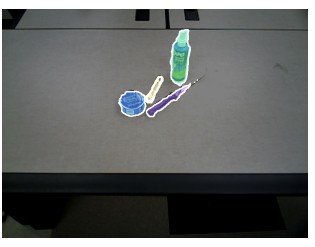

*RGB Image from Fetch*     *SAM Segmentation*     *UCN Segmentation*

**Scene-4**

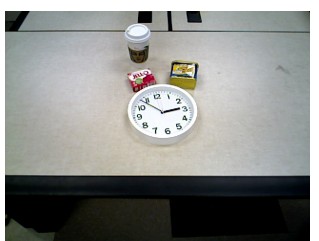 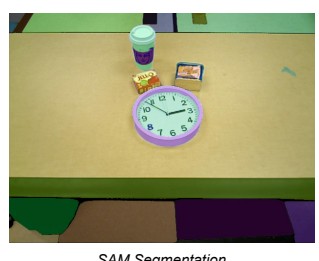 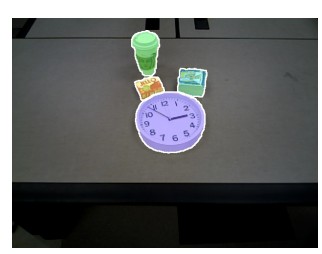

*RGB Image from Fetch*     *SAM Segmentation*     *UCN Segmentation*

**Scene-5**

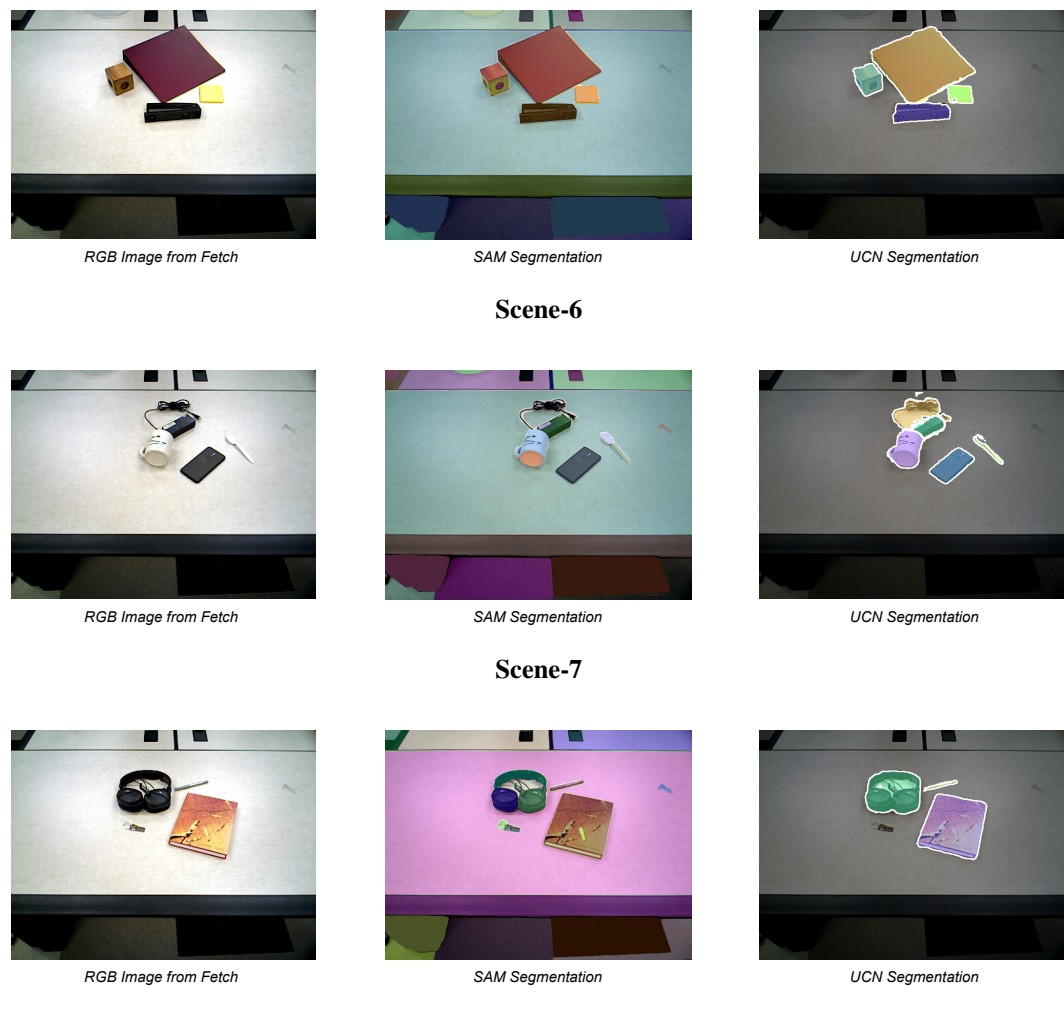

Figure 7: 8 real world scenes as used in FEWSOL (P et al., 2023). SAM (Kirillov et al., 2023b) predicts masks for the entire image in the absence of prompts like points or bounding boxes. Corresponding UCN (Xiang et al., 2021a) predictions which are far better in targeting object-centric masks.

