# OpenReview forum: "Proto-CLIP: A Vision-Language Prototype Alignment Approach for Few-Shot Learning"
_ICLR.cc/2024/Conference — ICLR 2024 Conference Withdrawn Submission_

### Official Review · Reviewer_xatT · 2023-10-30

**Soundness:** 2 fair
**Presentation:** 3 good
**Contribution:** 2 fair
**Rating:** 3
**Confidence:** 4

**Summary:**

In this paper, the authors introduce Proto-CLIP, which combines CLIP and prototypical networks to solve the problem of few-shot learning. Specifically, Proto-CLIP calculates the prototypes of the text and image features in the support set and calculates the similarity/distance of the data in the query set to the visual and textual prototypes in inference. Two variants of Proto-CLIP, i.e., training-free and fine-tuned versions, are proposed. The authors also propose to align the visual prototypes and textual prototypes of the same class to enhance performance. Extensive experiments are conducted on few-shot classification benchmarks and a real-world application on Robotics for this scenario is shown.

**Strengths:**

1. This paper is well-organized, with a clear and straightforward presentation of the motivation and methodology.
2. The authors conducted extensive experiments, covering a wide range of benchmarks. The authors also show real-world applications on robotic systems to verify the effectiveness of the proposed approach. Both qualitative and quantitative results are well demonstrated.
3. Proto-CLIP demonstrates superior performance compared to prior work across multiple benchmarks and settings.

**Weaknesses:**

1. The technical contributions of this work are somewhat limited. Firstly, CLIP is already a powerful zero-shot learner, as demonstrated in Table 4, achieving good performance without any adaptation. Secondly, the use of prototype networks, as mentioned in the introduction, is a well-established approach in few-shot learning. Therefore, the simple combination of prototype networks and CLIP may lack novelty or innovation and provide limited insights.
2. The authors mentioned that this work utilizes CLIP because of its powerful feature representation. Besides CLIP, there exist other power visual foundation models like DINO and DINO-v2, so why not verify these models?
3. Lack of in-depth analysis in some experiments: a) among all the tested benchmarks, what are the domains? How often do the concepts in these benchmarks appear in the pretraining data of CLIP and how does this affect the performance? b) An exploration of the effects of using only the text-encoder and only the image-encoder to compute the prototypes would provide valuable insights.
4. Some details in the paper require further clarification. In terms of image and text prototype alignment, Eq(4) shows that the prototypes are the mean of CLIP embeddings; Fig 2 shows that the CLIP encoders are frozen; Eq(5) defines the loss of alignment, and only the calculated prototypes are involved. This raises questions about the parameters responsible for learning text-image prototype alignment, considering the CLIP encoder parameters are frozen.

**Questions:**

Please refer to weakness, plus the following:

Q1: According to weakness 3-a), medical image benchmarks are not included in the evaluation, how does Proto-CLIP perform under this domain?

Q2: Refer to Sec 4.1 - "Backbones" discussion part, how to define better backbones?

Q3: Prototypes are simply calculated by taking the mean of all image/text features, are there any other variants that can weigh more on import samples and enhance proto-CLIP?

---

### Official Review · Reviewer_ycyz · 2023-10-30

**Soundness:** 2 fair
**Presentation:** 3 good
**Contribution:** 3 good
**Rating:** 3
**Confidence:** 4

**Summary:**

Authors propose a Proto-CLIP method that utilizes CLIP image and text encoders followed by designing a method to extract prototypes while learning adaptors and memory for few-shot classification. Their method allows for fine-tuning the learned memory initialized form support set. Additionally, they use contrastive image-to-text and text-to-image losses during training, unlike previous works that relied more on visual information solely.

**Strengths:**

- Good detailed analysis of their results in terms of ablations and comparison to SOA.
- Increasing the datasets with a new dataset FewSOL + the real world robotics setup.
- incremental but interesting novelty

**Weaknesses:**

- You need to clarify the memory how is it constructed there is a mention of using the support set to initialize the image memory and text memory. But it is not clear how they are fine-tuned and why they are not the same number as the prototypes are they for each shot in the K-shot all these need to be clarified in the Learning memories and adapter section.

- CoCoOp is missing from the comparison to SOA although it is better than their method in some datasets.
Zhou, Kaiyang, et al. "Conditional prompt learning for vision-language models." Proceedings of the IEEE/CVF Conference on Computer Vision and Pattern Recognition. 2022.

- “Therefore, we select the best configuration on the adapter and learnable text memory for each dataset in the following experiments. “ This indicates an unfair comparison to the state of the art as this means it is not just simply one model. It indicates the model itself changes from one dataset to another. Except that other SOA methods don’t do that, they do tune hyperparams and architectural choices but use the same across all datasets.

- Table 5 shots ablation gives the impression that their method on Imagenet is mostly on-par to TIP. Thus, the benefit from their method is quite not clear.

- FewSOL benchmark doesn’t seem to be an established benchmark for robotics. It is not clear what is its difference to other established benchmarks listed here:
—> Jiang, Huaizu, et al. "Bongard-hoi: Benchmarking few-shot visual reasoning for human-object interactions." Proceedings of the IEEE/CVF Conference on Computer Vision and Pattern Recognition. 2022.
—> Lomonaco, Vincenzo, and Davide Maltoni. "Core50: a new dataset and benchmark for continuous object recognition." Conference on robot learning. PMLR, 2017.
—> Massiceti, Daniela, et al. "Orbit: A real-world few-shot dataset for teachable object recognition." Proceedings of the IEEE/CVF International Conference on Computer Vision. 2021.
—> Siam, Mennatullah, et al. "Video object segmentation using teacher-student adaptation in a human robot interaction (hri) setting." 2019 International Conference on Robotics and Automation (ICRA). IEEE, 2019.

Although first two not exactly on fewshot but on related tasks why not simply evaluate on them? Or the third  and fourth ones are on fewshot and the last is even within a robotics task. What is the robotics aspect in FewSOL they are mentioning bringing to the problem?

**Questions:**

- What is the robotics aspect in FewSOL they are mentioning bringing to the problem?
- Does the method work on base and novel classes or is it only novel classes? If yes then what's the method performance on base classes? If no it should be stated clearly as a limitation w.r.t CoOp and others.

---

### Official Review · Reviewer_Gen4 · 2023-11-02

**Soundness:** 3 good
**Presentation:** 3 good
**Contribution:** 2 fair
**Rating:** 5
**Confidence:** 5

**Summary:**

This paper proposes to learn both image prototypes and text prototypes using CLIP for few-shot learning. The classification score between query image and support class is the weighted average score of the query<-->support visual prototype and query <--> support text prototype. The contrastive losses are used to align the image and text prototypes of the corresponding classes. Experiments are conducted on both few-shot learning benchmarks and real-world robot perception.

**Strengths:**

1. The idea of combining visual prototypes and textual prototypes for few-shot learning is reasonable and very interesting.

2. The experimental evaluations are sufficient with both few-shot learning benchmarks and real-world robot perception.

**Weaknesses:**

1. Some part of the model component is not clear. In Figure 2, how to generate the text prompts is not clear. In general, the prompts should be diversified for each of image-text pairs. Did the authors use the CoCoOp model?

2. The novelty is somehow limited as a lot of work have used both few-shot visual information and language semantic information for few-shot learning. e.g., [1,2]. The difference seems to be only the backbone.

[1]. Xing, Chen, Negar Rostamzadeh, Boris Oreshkin, and Pedro O. O Pinheiro. "Adaptive cross-modal few-shot learning." Advances in Neural Information Processing Systems 32 (2019).

[2]. Kaul, Prannay, Weidi Xie, and Andrew Zisserman. "Multi-Modal Classifiers for Open-Vocabulary Object Detection." arXiv preprint arXiv:2306.05493 (2023).

3. The proposed model is not robust and the hyper-parameters are carefully tuned for each of the dataset. This hurts the generalizability of the model.

4. Can the authors provide more explanations about why the model performs worse on 1shot, compared to previous works in Table 6. Ablation studies of each of the two modalities are missing (including vision-only and language-only).

**Questions:**

Please see the weaknesses above